# Factor productivity impacts of climate change and estimating the technical efficiency of cereal crop yields: Evidence from sub-Saharan African countries

**Ferede Mengistie Alemu** *, **Yismaw Ayelign Mengistu, Asmamaw Getnet Wassie**

Lecturer Department of Economics, Debre Tabor University, Bahirdar, Ethiopia

* ferededtu2030@gmail.com

**Data Availability Statement:** All relevant data are within the manuscript and its Supporting Information files.

## Abstract

The world aims to ensure environmental sustainability and consolidate agricultural factor productivity, yet the excruciating impact of climate change coincides and remains a persistent threat. Therefore, the study aims to estimate the technical efficiency of cereal crop yields and investigate the impacts of climate change on agricultural factor productivity. For this purpose, panel data from 35 sub-Saharan African countries between 2005 and 2020 was employed. For analysis, the pooled OLS and stochastic frontier models were employed. The results revealed that in the region, the average efficiency score for producing cereal crops between 2005 and 2020 was 83%. The stochastic frontier model results showed that labour contributed 51.5% and fertilizer contributed 5.7% to raising the technical efficiency of cereal crop yields, whereas arable land per hectare reduced the technical efficiency of cereal yields by 44.7%. The pooled OLS regression result showed that climate change proxies ($CO_2$ and methane emissions) diminish land, labour, and fertilizers productivity at a 1% significance level, whereas GDP per capita boosts significantly the total factor productivity in agriculture. This confirmed how climate change reduced land, labour, and fertilizer input productivity. The results concluded that the region had a high level of technical efficiency; of which labour and fertilizer inputs contributed the largest share; however, their productivity has dwindled due to climate change. To increase cereal crop yield efficiency and limit the adverse effects of climate change on agricultural input productivity, the region should combine skilled and trained labour and fertilizer with sophisticated agriculture technologies, as well as adopt climate resistance technologies (weather- resistant variety seed and planting revolution mechanisms).

## 1. Introduction

Agriculture is the backbone of a region's economy; however, its agricultural productivity growth has lagged behind that of Asia, America, and Europe [1]. Sub-Saharan Africa's average agricultural TFP growth rate was 0.14% per year between 1960 and 1984, but grew to 1.24%

**Funding:** The author(s) received no specific funding for this work.

**Competing interests:** the authors have no any computing interest

per year between 1985 and 2002. During this time, the average annual growth rate was around 0.6%, accounting for 36% of the overall rise in crop output [2]. With technology adoption, Africa's annual productivity growth rate is 3.13% [3]. Agricultural TFP growth is reshaped by education, capital intensity, and irrigation [4]; and improved institutional quality, less restrictive trade policies, better access to credit for the private sector, and improved connectivity between landlocked and non-landlocked nations [5].

By 2050, climate change will offset the improvements in agricultural productivity and its contribution to GDP in Africa. Especially in regions where irrigation is impeded, climate change is highly likely to provoke GDP losses [6, 7]. Even if $CO_2$ emissions and temperature consequences are expected to worsen, crop yields have already fallen in many parts of the world. Thus, by 2050, ensuring food security for nine billion people will be a complex task that requires organized and extensive adaptation activities [8]. However, $CO_2$ increases world yields by roughly 1.8% with the development of technology, while without technology, they decrease by 1.5% every decade [9]. Carbon dioxide ($CO_2$), nitrogen oxide ($N_2O$), and methane ($CH_4$) are the most prevalent greenhouse gases (GHGS), and they are creating a variety of serious consequences for agriculture [10, 11].

Food insecurity is spreading across Sub-Saharan Africa (SSA), where 123 million people (12 percent of the SSA's population) are expected to be acutely food insecure in 2022, suffering from severe malnutrition as well as being unable to achieve basic consumption needs [12, 13]. Climate change, which comprises variations in rainfall, drought, flooding, and warmer and cooler temperatures, has an adverse effect on food systems, ranging from direct effects on crop production to changes in markets, food pricing, and the supply chain [14]. Its impact negatively influenced the productivity of agricultural businesses and industries [15]. However, its impact is uneven worldwide because its severity is much wider in industrialized countries than in LDCs [16–19], plus its effect varies between the short run and the long run [20]; the consequences will be greater on agricultural factor productivity and environment unless the adoption options are held [21, 22]. In the region, it aggravated food insecurity by eroding the returns of agricultural production. However, climate-smart agriculture adoption has increased productivity, ensured environmental quality, and enhanced household food security [23–26]. Climate models forecast that agricultural productivity will be significantly reduced in the future. The expected rise in average global temperature induced by higher greenhouse gas (GHG) emissions into the atmosphere, as well as greater depletion of water resources due to increased climate variability, constitute a serious threat to global food security [27].

Agriculture in underdeveloped nations is vulnerable to climate change, and smallholder farmers have limited capacity to adapt to climate resilience technology [28]. In the short run, excessive heat has deteriorated agricultural input productivity. But climatic adaptations alleviate the problem by making flexible adjustments in labour, fertilizer, and machinery returns in the long run [29]. Crop diversification, shifting planting timings, and crop rotation/mixed cropping are the prime climate-smart agriculture practices used by farmers in Africa [30–34]. Farmers likelihood of being food secure and adopting climate-smart technology is strongly associated with their better education level, access to land, higher household income, and enough market access [35, 36].The expansion and effective dissemination of new agricultural techniques and technologies have enhanced crop productivity [37], but many households are reluctant to adopt technologies in reaction to climate change [33]. The impact of climate change has been regionally hampered by implementing new legislation and offering training to farmers about how to maximize technological efficiency in agriculture [38].

In Africa, CSA was embraced as a strategy to address concerns about agricultural productivity, improve adaptation measures, and increase climate resilience. However, in most countries, it has been challenged by a lack of national Climate-Smart Agriculture Investment Plans

(CSAIPs) [39], a lack of clear conceptual understanding, inadequate policies, and insufficient capital [40, 41]. It has also been affected by high initial investment costs, labour requirements, and management intensity associated with conservation agriculture and rainwater harvesting [42], gender gaps, ecological and environmental factors [43, 44], shortage of cropland, land tenure issues, lack of adequate knowledge about CSA, slow return on investment, and insufficient policy and implementation schemes [45, 46]. Climate-smart agriculture has been mainstreamed into agricultural development plans through the construction of regional, subregional, and national climate change policies and strategies aimed at mitigating climate change and strengthening African people's adaptive capacity [47]. Smallholders face challenges in implementing climate-smart agriculture (CSA) due to unclear duties, insufficient links between administration levels, limited resources, and political intervention [48]. African governments face political, economic, and governance problems while implementing climate-smart agriculture (CSA) to achieve the United Nations Sustainable Development Goals [49]. In most African countries, there is a lack of consideration for the replacement, complementing, or conditional effects of policy initiatives on the adoption of smart climate practices [50]. The government and policy alternatives for climate smart adoption have inadequate and ineffective interventions in decision making, accessing inputs, and formulating and implementing policies in Africa [51].

Literature examines the climate change effects on agricultural yields, particularly crop outputs. Land productivity has boosted crop yields when there is extensive adoption of technology and smart climate agriculture; however, this is not only due to rising polluted land size [52]. The returns on land size varies in small-scale farming and large-scale farming [53, 54], but constant land size decreases wheat yields where there is bad climate condition [55]. However, in well-developed African countries, high dependency of food on cereal crops increase the productive capacity of land [56]. In LDCs, the returns of agricultural inputs are stumpy in manipulating the growth of agriculture due to climate change, yet producers employ cropping intensity to compensate for falling average yields per harvest and to ensure sustainable land productivity development [57, 58]. Nonetheless, the productivity of inputs heavily depends on the farmers likelihood to adopt technology and returns to scale [59]. Improving labour skills through promotional initiatives as well as farmer training is a panacea for boosting crop productivity in developing countries [60], because an increase in farmers' perceptions of temperature advanced farm technical efficiency in the seeding and vegetative periods in both the constant and variable returns to scale models, unless temperature increases during crop harvesting had a negative effect on production efficiency [61, 62]. Manure application had a significant and positive impact on relative yield by boosting SOC storage and soil nutrients [63]. However, abuse and misuse of chemical fertilizers in agro-ecosystems can cause land degradation and lower productivity [64]. Organic fertilizer treatments resulted in an 11% -13% yield increase and a 4%-5% higher net economic gain [65, 66]. Evidence showed that labor productivity in farming is significantly affected by remittance [67], research and development [1], and the likelihood of being a part-time worker [68].

Environmental subsidies, especially rural subsidies, had a positive effect on technical efficiency and crop production [69]. Farmers have enormous potential to increase their output without using additional inputs when the land is virgin; yet, giving options is critical for boosting food supply in locations where virgin soil for food production is limited [70]. The technical efficiency score among countries varies due to the economic, institutional, environmental, and social complexity of agricultural productivity [71, 72]. Climate change undesirably affects the technical efficiency of crop yields by diminishing the marginal productivity of inputs in the region [73]; however, the technical efficiency of crops can be boosted by using existing resources and introducing new technology [74]. Agricultural production has improved at a

constant pace of 0.44% per year due to technical innovation and a change in input efficiency [58]. Furthermore, farm households that used high-yielding seed varieties and research-based production procedures beat their competitors in terms of technological efficiency [75]. With a technological efficiency of 38.2%, Africa is predicted to miss out on around 62% of its agricultural potential [76], particularly in Ethiopia, where the estimated TE of 66% implies possibilities for greater crop production efficiency [77].

Studies like [30–34] focused on the impact of climate change on crop yields, whereas [37, 60, 62, 73] investigated the impact of climate change on technical efficiency. However, they failed to investigate the impact of climate change on agricultural factor productivity as well as estimate the technical efficiency of crop yields in Sub-Saharan African countries. The study aims to address the following specific objectives: investigating the impacts of climate change on agricultural factor productivity, examining the impact of factor productivity on cereal crop yields, and estimating the level of technical efficiency of cereal yields in the region. The study was organized into introduction, results and discussion, and conclusion.

## 2. Methodology

### 2.1 Data source and definition of variables

For this study, the panel data have been employed and the data obtained from the World Bank data base (https://databank.worldbank.org/source/world-development-indicators). To analyze the data, the panel data estimation technique and stochastic frontier analysis were employed. The study focused on the production of the following cereal crops: wheat, rice, maize, barley, oats, rye, millet, sorghum, and buckwheat. The detailed definition of the data can be described in "Table 1."

**Table 1. Definition of variables.**

| Variables | Definitions |
|---|---|
| Agricultural methane (CO4) | *Agricultural methane emissions are emissions from animals, animal waste, rice production, and agricultural waste burning. It is negatively related to input productivity [10, 11].* |
| CO2 emission | *Carbon dioxide emissions are those stemming from the burning of fossil fuels and the manufacture of cement. It has negatively affected input returns [78– 81].* |
| GDP per capita income | *GDP is the sum of the gross value added by all resident producers in the economy plus any product taxes and minus any subsidies not included in the value of the products. It is directly related to input return [82].* |
| *Labor productivity $\left(\frac{y_{it}}{l_{ait}}\right)$* | *It is the percentage of total cereal yield, measured as kilograms per hectare of harvested land, per total labor engaged in agriculture. It is positively associated with the technical efficiency of crop yields [57, 67, 83, 84].* |
| cereal yield k.g/hectare | *Cereal yield, measured as kilograms per hectare of harvested land, includes wheat, rice, maize, barley, oats, rye, millet, sorghum, and buckwheat. It has been positively affected by labor and fertilizer but negatively affected by land size [57, 59].* |
| *Land productivity $\left(\frac{y_{it}}{l_{it}}\right)$* | *It is the percentage of total cereal yield, measured as kilograms per hectare of harvested land per total arable land per hectare. It is has been negatively associated with cereal crop efficiency [52, 53].* |
| *Fertilizer productivity $\left(\frac{y_{it}}{fe_{it}}\right)$* | *It is the percentage of total cereal yield, measured as kilograms per hectare of harvested land per total consumption of fertilizer in kilogram and it has a multiple effect on cereal crop efficiency [64, 83, 85].* |

Source: From literatures

## 2.2 Ethics approval and consent to participate

This article does not contain any studies with human participants performed by any of the authors.

## 2.3 Model specification

We developed the stochastic frontier model and the pooled OLS panel data model. We used the first model to assess the technical efficiency of cereal crop yields, and the second model to investigate the impact of climate change on labour, arable land, and fertilizer productivity. In the first model, the concept of technical efficiency is a central tool used to measure the firm performance. The stochastic frontier production function (parametric) and Data Envelopment Analysis (DEA) are the two most widely used nonlinear methods for evaluating efficiency/inefficiency. DEA has the ability to accommodate many outputs and inputs in technical efficiency analysis. Nonetheless, DEA fails to assess the potential influence of random shocks, such as measurement inaccuracy and other noise in the data [86].

Hence, stochastic frontier analysis is more applicable to analyzing the technical efficiency of agricultural production [87]. Thus, for this investigation, the stochastic frontier production function was used. The model was adopted from [19, 77, 88].The stochastic production frontier technique is best suited for efficiency studies that are likely to be influenced by situations beyond the control of the decision-making unit. This technique encountered challenges as a result of these elements, as well as technical defects that arise during measurement and observation [83].One component is a company's technological inefficiency, while the other is random shocks (white noise) caused by unfavorable weather, measurement errors, and variable omissions. The Cobb–Douglas production function of cereal yields in kilogram per hectare can be specified in (Eqs 1, 2 and 3).

$$Y_{it} = \left(x_{it}{}^{\beta} + e_i\right) \qquad \text{Eq 1}$$

$$lnY_{it} = ln\theta + \sum \beta_i x_{it} + exp^{ei} \qquad \text{Eq 2}$$

$$ei = vi - ui \qquad \text{Eq 3}$$

Where "$ln$" denotes the natural logarithm, "$it$" represent the $i^{th}$ countries cereal yield at time $t$ in the sample, "$Y_{it}$" represents the cereal yield in kilogram per hectare of $i^{th}$ countries at time $t$. "$x_{it}$" Refers to the farm inputs of the $i^{th}$ countries at time $t$, "$e_{it}$" $= v_{it} - u_{it}$ which is the residual random term composed of two elements $v_{it}$ and $u_{it}$. The $v_{it}$ is a symmetric component and permits a random variation in output due to factors such as weather, omitted variables, and other exogenous shocks.

The household level efficiency score is measured as the ratio of observed output to potential output is expressed in Eq 4.

$$TE_{it} = \frac{Y_{it}}{F(X_{it}, \beta)exp} = exp^{(-u)} \qquad \text{Eq 4}$$

"$TE_{it}$", Stands for the technical efficiency score of the $i_{th}$ countries, "$Y_{it}$" stands for the actual level of output, and "$F(Xi, \beta)exp$", stands for the potential level of output. The value of "$TE_{it}$", is limited within the range 0 to 1 [86]. The value 1 represents technical efficiency and value 0 is technical inefficiency.

To examine the factors affecting the household level inefficiency/efficiency score, the following model is established in Eq 5.

$$U_{it} = \alpha_0 + \beta X_{it} + e_{it}$$

Eq 5

*Where $U_{it}$ refers to the technical inefficiency/efficiency of countries, `"$\alpha_0$&$\beta$" are the parameter to be estimated,"$X_{it}$" refers to economic factors.*

In the second model, we have examined the impacts of climate change on agriculture factor productivity such as land, labor and fertilizer, the pooled OLS panel model was employed. Total factor productivity encompassed the multiple factor inputs in production, and it measure the productivity of firms over time [89]. Total factor productivity is drive from the Cobb–Douglas production function that describes in constant rate of return expressed in Eq 6.

$$Y_{it} = Ae^{yit}$$

Eq 6

Where, "$y_{it}$", the production function as a function of labor, land and fertilizer. The above equation rewrite as follows and it illustrated in Eq 7.

$$\log \frac{V_{it}}{L_{it}} = \alpha + (\alpha + \beta - 1)\log L_{it} + \beta \log \frac{K_{it}}{L_{it}} + \gamma_{it} + \delta \log \frac{F_{it}}{L_{it}} + \delta_{it} + U_{it}$$

Eq 7

Where, v,k,l &x, are value add, land, and fertilizer per labor ratio, respectively.

The general formula of factor productivity is described in total output per inputs. This can be expressed in Eq 8

$$A_{it} = \frac{Y_{it}}{x_{it}}$$

Eq 8

*Where, "$A_{it}$", inputs productivity (labor, land, and fertilizer inputs), "$y_{it}$" represent the $i^{th}$ countries cereal yield at time t, and "$x_{it}$", the $i^{th}$ countries inputs (labor land and fertilizer) at time t.*

The pooled OLS model is more applicable in the case of absence of individual heterogeneity. Furthermore, the Pooled OLS model effectively treats the panel data as cross-sectional data by ignoring entity-specific effects. It is commonly applied in the absence of hetroscdasticity or individual heterogeneity problem [90]. The LM test of heterogeneity results verified no individual heterogeneity. The model can be specified in Eq 9, Eqs 10 and **11**.

$$\frac{y_{it}}{l_{it}} = \alpha i + \beta co2_{it} + \theta methne_{it} + \gamma GDP_{it} + e_{it}$$

Eq 9

$$\frac{y_{it}}{la_{it}} = \alpha i + \beta co2_{it} + \theta methne_{it+\gamma GDP_{it}} + e_{it}$$

Eq 10

$$\boldsymbol{\frac{y_{it}}{fe_{it}} = \alpha i + \beta co2_{it} + \theta methne_{it+\gamma GDP_{it}} + e_{it}}$$

Eq 11

The above models satisfy the assumption of no individual heterogeneity. This can be illustrated in Eq 12.

$$cov(\alpha i, e_{it}) = 0$$

Eq 12

*Where ($l_{it}$), the land productivity for $i^{th}$ country at "t" time,($l_{ait}$) the labor productivity for $i^{th}$ country at "t"time, and ($fe_{it}$) the fertilizer consumption productivity for $i^{th}$ country at "t" time*

*period, CO2 and Methane are the proxy variables for climate change whereas αi Individual heterogeneity.*

## 3. Results and discussion

### 3.1 The average cereal crop yields and the level of technical efficiency in the region

Results show that, with homogeneous technology, the estimated mean technical efficiency of 38.2% implies that, in Africa, almost 62% of the potential agricultural output is untapped [76]. Cereal production in Sub-Saharan Africa has experienced low and unpredictable growth rates due to its rain- fed nature and subsistence orientation over the last four decades. In recent years, crops were planted on 98.6 million hectares, yielding 162 million tons [91]. In this study, Fig 1 sketches the average cereal crop yields and levels of technical efficiency in Sub-Saharan African countries. Between 2005 and 2020, countries that induced more than 1100 k.g of cereal yields had a 60% production efficiency score. This confirmed that countries that can harvest cereal yields of at least 1100 kilograms and up to 1600 kilograms can achieve production efficiencies of 60% and 90%, respectively. This proves that the region accounts for a minimum efficiency score of 0.6 (60%) and a maximum efficiency score of 0.9 (90%), with the lowest average cereal yields of 1100 kilograms and the highest yields of 1600 kilograms from 2005 to 2020. This has led to the conclusion that if agriculture is combined with high technological adoption, it can increase productivity and considerably reduce malnutrition in the region.

### 3.2 Cereal crop yields and factor productivity in Sub-Saharan African countries

In LDCs, the impacts of agricultural inputs are stumpy in influencing the growth of agriculture, but producers use cropping intensity to compensate for the dropping average yield per harvest and to assure sustainable land productivity development [57]. Organic fertilizer treatments resulted in an 11% -13% yield increase and a 4% -5% higher net economic gain [65, 66]. After the 2000s, the average annual growth rate of agricultural productivity is expected to be

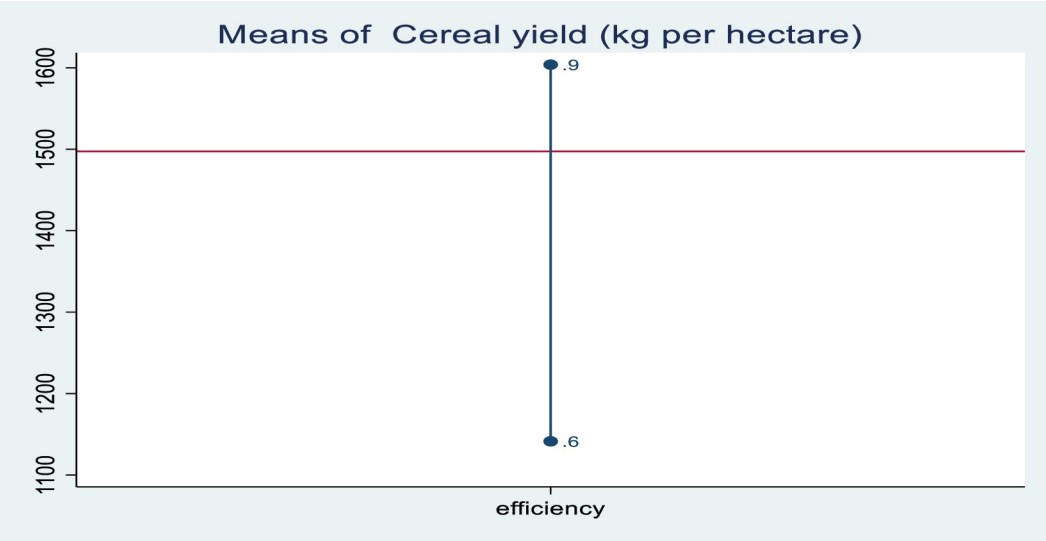

**Fig 1. Average cereal yields and the level of technical efficiency in the region.** Source: own computation STATA 17.

3.13% for the sample counties [3]. Fig 2 shows the trends of cereal crop yields in kilograms per hectare. The findings indicate that cereal crop yields in kilograms per hectare and input productivity remained steady between 2005 and 2020. Unlike Togo, which had a high level of fertilizer productivity between 2005 and 2010, the region experiences comparable long-term fluctuations. This is due to insufficient investment in farm-level research, harsh geographical conditions and disputes, a failure to absorb production and marketing risks, and ineffective incentives. As a result, in order to boost land, labour, and fertilizer productivity, technical innovation must be pursued until the economy is able to ensure food security through the spread and acceptance of smart agricultural technology.

### 3.3 Climate change and cereal crop yield in sub Saharan African countries

Fig 3 shows the link between climatic change and cereal crop yields in kilograms per hectare in SSA countries. With the expansion of technology, CO2 increases world yields by roughly 1.8% every decade while decreasing them by 1.5% every decade [9]. Carbon dioxide ($CO_2$), nitrogen oxide ($N_2O$), and methane ($CH_4$) are the most prevalent greenhouse gases (GHGS), and they are creating a variety of serious consequences in the agriculture sector [10, 11]. In this study, climate change and cereal yields show a consistent and similar pattern in the majority of the region from 2005 to 2020. However, numerous Sub-Saharan African countries, including South Africa and Angola, have high carbon emissions and low cereal crop yields; in particular,

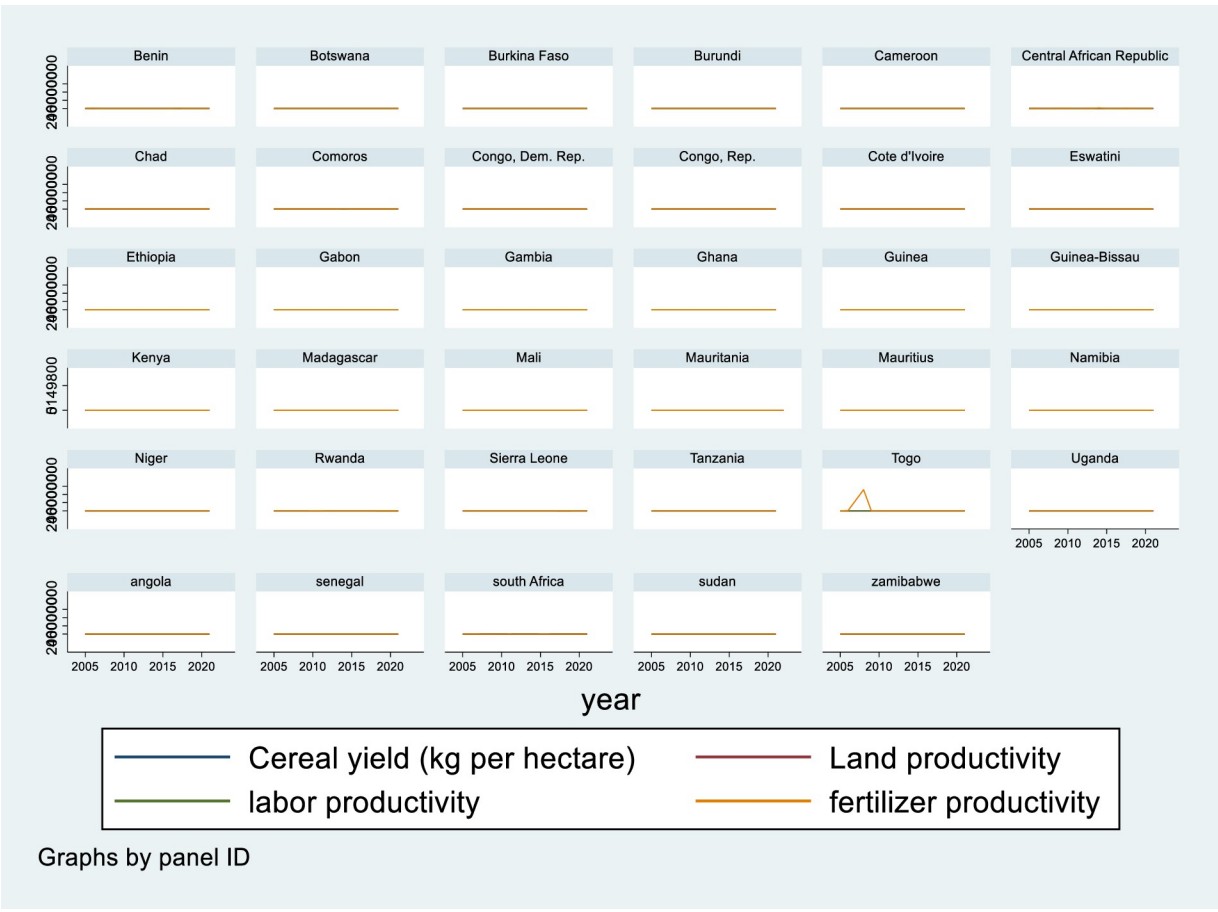

**Fig 2. Cereal crop yields and factor productivity in sub-Saharan African countries.** Source: own computation STATA 17.

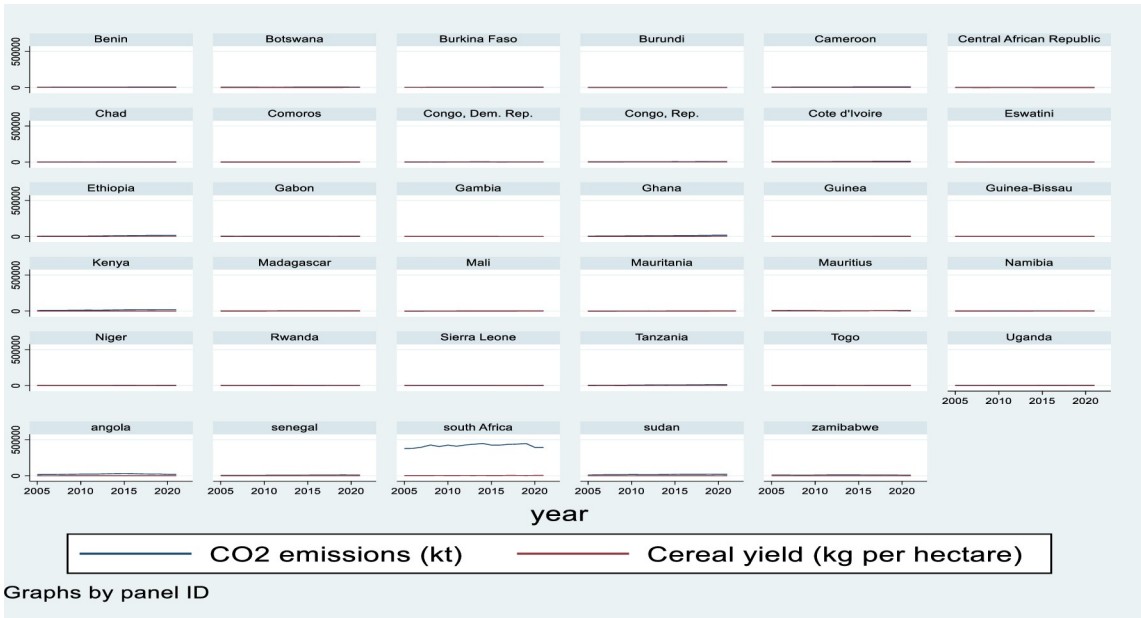

**Fig 3. Climate change and cereal yield in sub Saharan African countries.** Source: own computation STATA 17.

South Africa's carbon emissions are disproportionately high in relation to its cereal crop yields.

### 3.4 The technical efficiency scoring of cereal crop yields in the region

A firm is said to be technically efficient if it produces the most output with the least amount of inputs, such as labor, capital, and technology. Technical efficiency refers to how well a company or system maximizes production using a limited amount of inputs. With a technological efficiency of 38.2%, Africa is predicted to miss out on around 62% of its agricultural potential [76], particularly in Ethiopia, where the estimated TE of 66% implies possibilities for greater crop production efficiency [77]. A firm is considered technically efficient if it is able to manufacture more goods without expanding the number of production inputs. We have investigated and estimated the technical efficiency score of cereal crop production in the region. The results in Fig 4 demonstrate that the region has an average minimum technical efficiency score of 0.683 and a maximum technical efficiency score of 0.994. This signifies that the region is highly technologically efficient.

### 3.5 The analysis of the descriptive statistics and correlation coefficients in the region

From 1960 to 1991, crop output in SSA increased by 2.7% each year, while food production increased by 2.4% per year. But, worker productivity declined by an average of one percent per year in SSA agriculture. Crop productivity dropped between 2008 and 2019, with no evidence of improvement [92]. "Table 2" shows the average cereal crop yield per unit of input. The results confirmed that the average cereal crop yield per hectare from 2005 to 2020 was 1497.332kg. During these years, 99% of the countries in the region have produced a maximum of 7541kg per hectare. However, the productivity of inputs such as labour, land, and fertilizer use has had different effects on cereal crop yields across countries in the region.

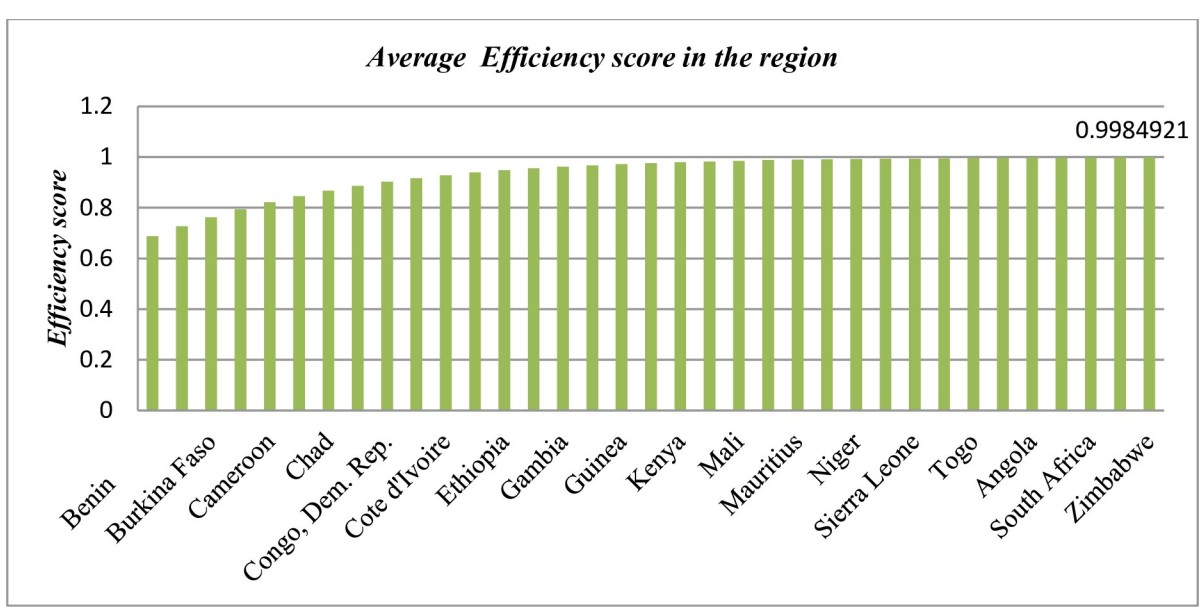

**Fig 4. The technical efficiency scoring of cereal crop yields in the region.** Source: own computation STATA 17.

Over these periods, the average productivity of fertilizer adoption, labour, and land was 10125.08, 0.001, and 0.004 kg, respectively. It was revealed that, as compared to fertilizer productivity, expanding harvested area diminished average returns on cereal crop yields in agriculture. This analysis confirmed that fertilizer use constituted nearly all the cereal crop yields from 2005 to 2020 in Sub-Saharan African countries, whereas land and labour inputs had virtually no impact. From 2005 to 2020, this study confirmed that fertilizer adoption contributed the most to the growth of cereal crop yields in Sub-Saharan African countries, followed by labour and land. The marginal productivity of land and labour had decreased as a result of the region's agriculture sector's lack of innovation and slow adoption of technology. As a result, the region achieved an average technical efficiency of 83%, which is not particularly notable given that the minimum efficiency score is 60% and the highest efficiency score is 90%.

The correlation coefficient is the unit of measurement used to calculate the intensity of the linear relationship between the variables involved in a correlation analysis. It is easily identifiable because it is represented by the symbol "r" and ranges from 1 to -1. The negative value implies negative relations, whereas the positive value suggests positive relations between two variables. Agriculture in underdeveloped nations is vulnerable to climate change, and smallholder farmers have limited capacity to adapt to climate resilience technology [28]. In the short run, excessive heat has deteriorated agricultural input productivity. But climatic

**Table 2. Analysis of descriptive statistics.**

| Variables | Mean | Std. Dev. | Min | Max | p1 | p99 | Skew. | Kurt. |
|---|---|---|---|---|---|---|---|---|
| Ave. Efficiency score | 0.831 | .126 | 0.6 | 0.9 | 0.6 | 0.9 | -1.29 | 2.665 |
| Cereal yield kg/ hectare | 1497.332 | 1138.51 | 217.6 | 9453.7 | 316.6 | 7541 | 3.472 | 19.34 |
| Land productivity | .004 | .014 | 0.012 | .119 | 0.010 | .094 | 6.003 | 41.476 |
| labor productivity | .001 | .002 | 00.01 | .017 | 0.001 | .014 | 3.915 | 20.404 |
| fertilizer productivity | 10125.08 | 211000 | 2.703 | 5149800 | 6.58 | 41338.785 | 24.245 | 589.862 |

Source: own computation STATA 17

adaptations alleviate the problem by making flexible adjustments in labour, fertilizer, and machinery returns in the long run [29]. "**Table 3**" shows a negative correlation between CO2 emissions and factor productivity, while there is a positive correlation between factor productivity and cereal yields.

### 3.6 The panel unit root test by using fisher type test of unit root

For unbalanced and fixed panel data, the Fisher-type unit root test is fitted to free the model from spurious regression [93]. In the fisher-type unit root test, the null hypothesis represents the unit root, and the alternative is the stationary one. If the probability values are less than 5%, reject the null hypothesis, and the data is stationary [94]. The results in "Appendix 1" confirm that the data are stationary.

### 3.7 Analysis of the impact of climate change on agriculture input productivity

In "**Table 4**" diagnostic tests such as hetroscdasticity and serial autocorrelation tests have been checked by using the Breusch and Pagan Lagrangian multiplier tests and the Wooldridge autocorrelation test. The LM test is appropriate for testing the diagnostic test of any specified model; it assumes that the null hypothesis is designed with constant variance [93]. The Wooldridge autocorrelation test in panel data is the most commonly used test in panel data sets [95]. Results in this study verified the absence of autocorrelations and the constant variance (no hetrosckdasticity) problem.

"**Table 4**" showed the impact of climate change proxy (agricultural methane and CO2 emissions) on productivity of arable land, agricultural labor, and fertilizer adoption per kilogram. The results disclosed that agricultural methane (CO4) and CO2 emissions have a harmful and noteworthy impact on agricultural factor productivity in sub-Saharan African nations. Agricultural methane (CO4) and CO2 emissions have a weighty and adverse impact on labor, land, and fertilizer productivity. Studies reveal that agricultural methane and nitrous oxide emissions have a dangerous impact on agricultural factor productivity [80, 81, 96]. Moreover, CO2 emissions reduce household welfare and agricultural output by reducing both traded and non-traded food output [97]. Also, climate models forecast a progressive rise in carbon dioxide (CO2) concentration and temperature over the globe, adversely affecting agricultural factor productivity [79]. However, the growth of agricultural output enhances the quality of the climate by degenerating greenhouse emissions and is used to mitigate climate change crises [98, 99]. In poorer nations, a 1°C increase in yearly temperature reduces TFP growth rates by 1.1–1.8 percentage points, while its impact is negligible in richer countries [100].

Land productivity diminishes when the annual average maximum temperature rises. It is positively associated with irrigation area, pump set number, and fertilizer application per

**Table 3. Analysis of correlation matrix.**

| Variables | (1) | (2) | (3) | (4) | (5) |
|---|---|---|---|---|---|
| (1) Land productivity | 1.000 | | | | |
| (2) labor productive | 0.901 | 1.000 | | | |
| (3) fertilizer productivity | -0.010 | -0.008 | 1.000 | | |
| (4) Cereal yield kg/hectare | 0.728 | 0.577 | -0.014 | 1.000 | |
| (5) CO2emissionskt | -0.053 | -0.085 | -0.011 | 0.397 | 1.000 |

Source: own computation STATA 17

**Table 4. The impact of climate change on agriculture input productivity.**

| Variables | Model(1) | Model(2) | Model(3) |
|---|---|---|---|
| | **Land productivity** | **Labor productivity** | **Fertilizer productivity** |
| Agricultural methane | -5.80–08 (3.55–08) | -2.89–08 *** (5.77–09) | -.000018*** (4.57–06) |
| $CO_2$ emission | -5.13–08*** (7.68–09) | -8.16–09 *** (1.25–09) | 1.46–06 (9.89–07) |
| GDP per capita income | 3.39–06 (2.35–07)*** | 4.83–07 *** (3.82–08) | .0003466 *** (.0000303) |
| constant | -.0013092 (.0008151) | .0005729 (.0001325) | 6.012633*** (.1052361) |
| Mean dependent var | 0.004 | 0.001 | 10125.080 |
| Overall $R^2$ | 0.92 | 0.91 | 0.702 |
| $x^2$ | 243.763 | 243.100 | 1.328 |
| Number ofobservation | 595 | 595 | 594 |
| Prob $> x^2$ **** | 0.000 | 0.000 | 0.723 |
| | | | Breusch and Pagan Lagrangian multiplier test |
| Prob $>$ chibar2 | 1.00 | 1.00 | 1.00 |
| | | | **Wooldridge test for autocorrelation in panel data** |
| No Autocorrelation | F(1, 16) = 2.166 Prob $>$ F = 0.1605 | F(1, 16) = 3.665 Prob $>$ F = 0.0736 | F(1, 16) = 1.879 Prob $>$ F = 0.1894 |

Source: own computation STATA 17

Note: in the above regression models results (***, **&*) are represented the 1%, 5% and 10% statistically level of significance. Diagnostic tests have confirmed no serial correlation and non-constant variance.

hectare of land [72]. In model 1, results showed that agricultural methane and CO2 emissions have a negative impact on land productivity. A one percent release of agricultural methane and CO2 emission into the atmosphere reduces the average returns of land productivity for grain yields by 5.80 kg and 5.13 kg, respectively, in the region. Emissions of methane and CO2 cause to high level of flooding, soil erosion, acid rain, and drought, which lie-down the productivity of land. However, GDP per capita income has growing land productivity. Rising GDP per capita income has aided the agriculture industry by expanding various land management programs, fostering innovation, and promoting technological uses.

Studies [9, 101, 102] revealed that climate change has an adverse impact on fertilizer productivity and cereal yields. Methane and CO2 emissions also lead to the formation of ozone, which lowers air quality and causes different health problems in animals, premature human mortality, and reduced crop yields. Similar to the previous studies, climate change has a negative impact on fertilizer productivity and application. High carbon and methane emissions create high levels of rainfall and drought, which impede the application of fertilizer. Not only does climate change affect fertilizer application, but it also has consequences that go beyond the reduction of fertilizer production. Fertilizers are critical for feeding the world's rising population by enhancing crop yields and ensuring the sustainability of the environment. However, its consequences will be serious in the long run when countries apply the high-yield crop agriculture system. By boosting agricultural yields, we can lower the quantity of land required for agriculture. This can be ensured with the support of GDP.

Global climate change will increase outdoor and indoor heat loads and may impair health and productivity for millions of working people. Under the simple assumption of no specific adaptation, climate change will decrease labor productivity in most regions [103]. Current climate conditions already negatively affect labor productivity, notably in tropical countries [104]. The study found that climate change has global socioeconomic impacts, but its impact is very significant in high-temperature zones. Low- to medium-temperature zone variation has economic benefits by creating disparities in labor productivity among areas [82]. In this study,

climate change has a detrimental impact on worker productivity. Climate change has adverse effects on farmers' social and economic status by degrading the quality of health and education, as well as delaying labor's energy and profitable power, all of which contribute to the collapse of labour productivity in the agriculture sector. Flooding, corrosive rain, and the release of severe weather into the ecosystem all reduce input productivity during harvest. These serious difficulties brought on by climate change undermine workers' motivation to dedicate their resources and knowledge to cereal crop production. The quality of labor employed in the agriculture industry is low, with family labor having almost zero marginal productivity, and the impact of climate change has exacerbated labor productivity in sub-Saharan African countries' agriculture sectors.

In this study, GDP per capita income has a favorable and statistically significant impact on arable land productivity, labour productivity, and fertilizer productivity. The findings show that a one-unit increase in GDP per capita income leads positively to a noted one- unit increase in total factor productivity of land, labour, and fertilizer. GDP per capita growth has dramatically boosted overall agriculture factor productivity by enabling the private and public sectors to expand and embrace new farming methods, technology, and infrastructure that can boost these factors' productive capacity. It raises the marginal and average productivity of total factor productivity by developing infrastructure, technology, research and development, and applying structural innovation in Sub-Saharan African agriculture.

## 3.8 Analysis of technical efficiency cereal crop yields using the stochastic frontier model

"Table 5" shows the technical efficiency of cereal crop production using stochastic frontier analysis. Land fragmentation and small farm sizes are expected to be significant impediments for agricultural sector expansion. Nationwide plot-level data from Rwanda suggest a constant return to scale agricultural production function, as well as a strong negative link between farm size and output per hectare [105]. When labor and land are scarce, the size of the land has a favorable impact on the productivity of smallholder farmers. In this study, the stochastic frontier analysis results reveal that the elasticity of arable land, labor supply, and fertilizer adoption

**Table 5. Stochastic frontier analysis of the efficiency model.**

| ln cereal yield per hectare | Coef. | St.Err. | t-value |
|---|---|---|---|
| ln Arable land per hectares | -0.447*** | 0.036 | -12.55 |
| ln Labor in agriculture | 0.515*** | 0.039 | 13.09 |
| fertilizer consume per k.g | 0.057*** | 0.012 | 4.82 |
| Constant | 5.804*** | 0.223 | 26.02 |
| mu | 0.001509 | 0.002 | 0.90 |
| eta | .162*** | 0.034 | 4.79 |
| lnsigma2 | -1.491*** | 0.058 | -25.66 |
| lgtgamma | -44.308 | 0.001 | 0.0001 |
| Number of obs | 594 | | |
| Prob > chi2 | 0.000 | | |
| Chi-square | 287.167 | | |
| Akaike crit. (AIC) | 812.938 | | |

Source: own computation STATA 17

Note: in the above regression models results (***, **&*) are represented the 1%, 5% and 10% statistically level of significance.

has an extensive effect on the technical efficiency of cereal production. Thus, the elasticity of labor and fertilizer has improved the technical efficiency of cereal crops-; however elasticity of arable land size has decreased cereal crop efficiency. The findings suggest that increasing the elasticity of land size by one hectare reduces the efficiency of cereal crop output in (kilograms/ hectare) by 0.447 percent. However, a 1% increase in agricultural labour and a 1% increase in fertilizer consumption per hectare both contributed positively to a 0.515% and 0.057% rise in cereal crop production efficiency per hectare, respectively. Land size in the region has a negative impact on the efficiency of cereal crop output productivity because increased land size causes farmers to prioritize less in terms of adopting new technology, harvesting on a regular basis, and improper land management.

Chemical fertilizer adoption has a positive impact on the efficiency of crop productivity in the short term, but it has a negative impact on environmental sustainability and crop productivity in the long run [85]. It contributes to soil fertility and productivity when combined with more organic manure [106]. In this study, chemical fertilizer consumption increased crop output in the short run, despite its negative effects on environmental sustainability and land productivity. Furthermore, in the current circumstances, an increase in skilled labor has favorably boosted cereal crop output, implying that trained labor is necessary to embrace new methods of harvesting in agriculture. This has led to the conclusion that the use of skilled labour and fertilizer consumption in crop production increases crop production efficiency in the region, and their contribution expands to the adoption of smart technology, ensuring environmental sustainability and the future of billions of Africans.

## 4. Conclusions and policy implications

### 4.1 Conclusions

Today, the world is confronted with climate change, and its threat coexists after all notable mitigation strategies have been enacted by governments and organizations'. This is harming the agriculture industry and raising a sense of food insecurity, and its consequences have been the main driver of crop production inefficiencies in Africa. This study investigated the effects of climate change on agricultural factor productivity and estimated the technical efficacy of cereal crop yields in Sub-Saharan African countries. The pooled OLS and stochastic frontier models were used, with panel data sets ranging from 2005 to 2020. It was revealed that, regardless of Togo's tremendous consumption of chemicals, factor productivity and cereal crop production in all Sub-Saharan African countries complied with an analogous pattern from 2005 to 2010. From 2005 to 2020, most Sub-Saharan African countries experienced an analogous progression in climate change and cereal crop yields in kilograms per hectare, but South Africa and Angola suffered catastrophic climate change. The region had an average technical efficacy of 83%, with a maximum of 99.4% and a minimum of 68.3%. Labour and land have low contributions; however, fertilizer consumption has notable contributions to the technical efficiency of agricultural crops in the region. Also, climate change adversely affected the productivity of inputs. It confirmed how climate change adversely affected land labour and fertilizer productivity in Sub-Saharan Africa. Because acidic rainfall, flooding, and land erosion caused by climate change have compacted the marginal efficiency of land productivity, it also potentially diminishes the demand for working in agriculture and dampens farmers' desire to adopt new technology in agriculture while maximizing crop yields. Furthermore, it has a harmful effect on fertilizer productivity by raising its relative expenses and decreasing its relative return on crop yields. Rising GDP per capita income has boosted agricultural input productivity by expanding various land management programs, encouraging innovation, and promoting technological adoption. Fertilizers growth rate and labour growth rate both contribute positively to

achieving the maximum growth rate of technical efficiency of cereal crop yields, but land size has the opposite effect. The study concluded that CO2 and agricultural methane emissions reduce labour, land, and fertilizer productivity, but GDP per capita income increases factor productivity.

## 4.2 Policy implications and research limitation

The study has the following policy implications: To increase cereal crop yield efficiency and limit the negative effects of climate change on agricultural input productivity, the region should increase labour skills, adopt fertilizer with sophisticated agriculture-based technologies, and adopt climate resistance technologies (weather- resistant seed varieties and planting revolution mechanisms). Worldwide and nationwide governments and nongovernmental organizations should give high attention to mitigating the impact of climate change on input productivity. We suggested that future research should include the effects of technical efficiency and climate change on food insecurity in the region. Due to the absence of data, this study does not include all sub-Saharan African countries and other variables like improved seed and mechanical technology.

## Supporting information

**S1 Appendix.**
(DOCX)

**S1 Data.**
(XLSX)

## Acknowledgments

We acknowledged Debre Tabor university Economics and English Department staffs for their contribution to conduct this manuscript.

## Author Contributions

**Data curation:** Ferede Mengistie Alemu, Asmamaw Getnet Wassie.

**Formal analysis:** Ferede Mengistie Alemu.

**Methodology:** Asmamaw Getnet Wassie.

**Writing – original draft:** Ferede Mengistie Alemu.

**Writing – review & editing:** Ferede Mengistie Alemu, Yismaw Ayelign Mengistu.

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
