## [Decision Letter · Decision Letter 0]

14 May 2024

PONE-D-24-08690Factor productivity impacts of climate change and estimating the technical efficiency of cereal cropyields: Evidence from sub-Saharan African countriesPLOS ONE Dear Dr. ALEMU, Thank you for submitting your manuscript to PLOS ONE. After careful consideration, we feel that it has merit but does not fully meet PLOS ONE’s publication criteria as it currently stands. Therefore, we invite you to submit a revised version of the manuscript that addresses the points raised during the review process.

We look forward to receiving your revised manuscript.

Kind regards,

Fabien MUHIRWA

Academic Editor

PLOS ONE

Journal Requirements:

4. "We note you have included a table to which you do not refer in the text of your manuscript. Please ensure that you refer to Table 5 in your text; if accepted, production will need this reference to link the reader to the Table.

**Additional Editor Comments:**

Much criticism has been raised by reviewers, please try to address seriously and make your manuscripts more scientifically oriented interest. All the best.

Reviewers' comments:

Reviewer's Responses to Questions

**Comments to the Author**

1. Is the manuscript technically sound, and do the data support the conclusions?

Reviewer #1: No

Reviewer #2: Yes

2. Has the statistical analysis been performed appropriately and rigorously? 

Reviewer #1: No

Reviewer #2: Yes

3. Have the authors made all data underlying the findings in their manuscript fully available?

Reviewer #1: Yes

Reviewer #2: Yes

4. Is the manuscript presented in an intelligible fashion and written in standard English?

Reviewer #1: No

Reviewer #2: Yes

5. Review Comments to the Author

Reviewer #1: Dear Authors

In this manuscript the relationship between climate change and technical efficiency has been investigated. at first view, Its interesting, but in this version is not satiable for publishing. I recommend that you have to boost the problem statement, literature review and methodology. I have not seen any novelty in this paper. Please rewrite it again and resubmit.

Regards

Reviewer #2: Although the paper touches an important topic but there are few draw backs that should be tackle before acceptance:

In the abstract, results should be presented in quantitative terms.

It is better to present the situation of agriculture concerning climate change in Sub Sahara Africa in the Introduction section. It will be helpful to draw a research gap.

The literature review should be updated by considering each variable of the research model.

Please indicate the specification of all models that the author/s took into results. Why have they used different models?

It is better to indicate which cereal crop was taken for research and individually present their yield in a table instead of Figure 1.

Table formatting should be consistent throughout the paper.

Policy recommendations should be present after the conclusion part.

Research limitations should be present.

6. PLOS authors have the option to publish the peer review history of their article (what does this mean?). If published, this will include your full peer review and any attached files.

Reviewer #1: No

Reviewer #2: **Yes: **Nabila Khurshid

---

## [Author Response · Author response to Decision Letter 0]

1 Jun 2024

Response to Reviewer #1

Dear Reviewer One. We thank you for sharing your knowledge and skills, and your suggestions have greatly aided us in rewriting and enriching our work.

1. The abstract has been rewritten by considering the introduction, methodology, objectives, findings, conclusion, and policy implications. 

2. Additionally, it was suggested that the study's paper be rewritten and expanded to include unique content. We are dedicated to answering your question, and the gap is rewritten as follows: Studies like [5], [24], [27]–[29] focused on the impact of climate change on crop yields whereas [23], [36], [39], [40] investigated on climate change impact on technical efficiency. However, they failed to investigate the impact of climate change on agricultural factor productivity as well as estimate the technical efficiency of crop yields in Sub-Saharan African countries. The study aims to address the following specific objectives: investigating the impacts of climate change on agricultural factor productivity, examining the impact of factor productivity on cereal crop yields, and estimating the level of technical efficiency of cereal yields in the region.

3. The literature was edited and expanded to include recent related literature. We included literature for each variable and identified gaps. The methodology section was unclear, particularly the panel data model. We are now revising the model from specification to model formulation by providing a trustworthy reasoning for its application. The study's discussion, conclusion, and policy implications have all been updated. The updated section is indicated in red colour. 

4. We are free to receive comments and revised again

Response to Reviewer #2: 

Thank you for your time and important thoughts. 

1. The abstract should present quantifiable results. Here, the abstract is rewritten in quantitative terms “The world aims to ensure environmental sustainability and increase agricultural factor productivity, yet the harsh impact of climate change coexists and remains a persistent threat. Therefore, study aims to estimate the technical efficiency of cereal crop yields and investigate the impacts of climate change on agricultural factor productivity. For this purpose, panel data from 35 sub-Saharan African countries between 2005 and 2020 was employed. For analysis, the pooled OLS and stochastic frontier models were employed. The results revealed that in the region, the average efficiency score for producing cereal crops between 2005 and 2020 was 83%. The stochastic frontier model results showed that labour contributed 51.5% and fertilizer contributed 5.7% to raising the technical efficiency of cereal crop yields, whereas arable land per hectare reduced the technical efficiency of cereal yields by 44.7%. The pooled OLS regression result showed that climate change proxies (CO2 and methane emissions) diminish land, labour, and fertilizers productivity at a 1% significance level, whereas GDP per capita boosts significantly the total factor productivity in agriculture. This confirmed that agricultural methane emissions (CH4), carbon dioxide emissions (CO2) have reduced land, labour, and fertilizer input productivity. The results concluded that the region had a high level of technical efficiency; of which labour and fertilizer inputs contributed the largest share; however, their productivity has dwindled due to climate change. To increase cereal crop yield efficiency and limit the negative effects of climate change on agricultural input productivity, the region should combine skilled labour and fertilizer with sophisticated agriculture-based technologies, as well as adopt climate resistance technologies (genetically improved seed and planting revolution mechanisms).”

2. It is preferable to present the agricultural condition in Sub-Saharan Africa as part of the introduction. We believe that your comments are valuable, and we have added literature about the significance of agriculture and how climate change affects agriculture. For example, In Sub-Saharan African (SSA) countries, agriculture contributes 40% of GDP and employs more than half of the workforce [1]. By 2050, improvements in agricultural productivity in the region will significantly offset the GDP losses due to climate change. However, in regions where irrigation will be extended, climate change is insufficient to offset GDP losses [2] . You can see the revised, and the revised parts are highlighted in red color.

3. It will be helpful to draw a research gap. The research gap also rewrite and enriched like(Studies like [5], [24], [27]–[29] focused on the impact of climate change on crop yields whereas [23], [36], [39], [40] investigated on climate change impact on technical efficiency. However, they failed to investigate the impact of climate change on agricultural factor productivity as well as estimate the technical efficiency of crop yields in Sub-Saharan African countries. The study aims to address the following specific objectives: investigating the impacts of climate change on agricultural factor productivity, examining the impact of factor productivity on cereal crop yields, and estimating the level of technical efficiency of cereal yields in the region).

5. Update the literature review to reflect each variable in the study model. The literature was generated by incorporating relevant material for each variable. The literature part of the study contains references to CO4, CO2, labour, land, and fertilizer production. 

6. Please specify the models used in the results. Why did they utilize separate models? The reason for using various models is that the study has two objectives. The first goal is to examine the technical efficiency of cereal crops, and the second is to investigate the influence of climate change on input productivity. As a result, the stochastic frontier modal should be used to address the first objective, while one of the panel models can address the second. Based on the specification test of panel models, the Pooled is best model in this circumstance. The approaches are revised by justifying their reasoning using literature. See the revised, which are indicated in red color.

7. We have ensured consistency in table and figure formats, corrected issues, and reworked the entire presentation.

8. Policy recommendations and research limitations are offered as distinct subtitles following the study's conclusion.

 We notify you that the authors are eager to take any comments that the reviewer believes as they should be corrected further.

---

## [Decision Letter · Decision Letter 1]

12 Aug 2024

PONE-D-24-08690R1Factor productivity impacts of climate change and estimating the technical efficiency of cereal cropyields: Evidence from sub-Saharan African countriesPLOS ONE

Dear Dr. ALEMU,

Thank you for submitting your manuscript to PLOS ONE. After careful consideration, we feel that it has merit but does not fully meet PLOS ONE’s publication criteria as it currently stands. Therefore, we invite you to submit a revised version of the manuscript that addresses the points raised during the review process especially for the new reviewer raised comments.

We look forward to receiving your revised manuscript.

Kind regards,

Fabien MUHIRWA

Academic Editor

PLOS ONE

Reviewers' comments:

Reviewer's Responses to Questions

**Comments to the Author**

1. If the authors have adequately addressed your comments raised in a previous round of review and you feel that this manuscript is now acceptable for publication, you may indicate that here to bypass the “Comments to the Author” section, enter your conflict of interest statement in the “Confidential to Editor” section, and submit your "Accept" recommendation.

Reviewer #2: (No Response)

Reviewer #3: (No Response)

2. Is the manuscript technically sound, and do the data support the conclusions?

Reviewer #2: Yes

Reviewer #3: Partly

3. Has the statistical analysis been performed appropriately and rigorously? 

Reviewer #2: Yes

Reviewer #3: Yes

4. Have the authors made all data underlying the findings in their manuscript fully available?

Reviewer #2: Yes

Reviewer #3: Yes

5. Is the manuscript presented in an intelligible fashion and written in standard English?

Reviewer #2: Yes

Reviewer #3: Yes

6. Review Comments to the Author

Reviewer #2: (No Response)

Reviewer #3: This manuscript titled “Factor productivity impacts of climate change and estimating the technical efficiency of cereal crop yields: Evidence from sub-Saharan African countries” estimates the technical efficiency of cereal crop yields and examines the effects of climate change on agricultural productivity in 35 sub-Saharan African countries from 2005 to 2020. Using pooled OLS and stochastic frontier models, the average efficiency score for cereal production was found to be 83%. Labour and fertilizer significantly enhanced efficiency, contributing 51.5% and 5.7%, respectively, while arable land per hectare negatively impacted efficiency by 44.7%. Climate change indicators such as CO2 and methane emissions significantly reduced the productivity of land, labour, and fertilizers, whereas GDP per capita positively influenced overall agricultural productivity. The findings highlight high technical efficiency in the region, primarily driven by labour and fertilizer inputs, but these have been adversely affected by climate change. This work is an interesting contribution, and the authors are commended for their efforts. However, in order for the manuscript to meet the standards for publication, this submission requires a major revision. Hence, the following specific revisions are necessary:

1. The introduction discusses climate change impacts and the need for climate-smart agriculture in Sub-Saharan Africa (SSA) but fails to address the diverse local contexts that influence the adoption and effectiveness of these practices. Different countries and regions face unique climatic challenges, agricultural methods, and socio-economic conditions. Consequently, more nuanced research findings and examples from specific countries or communities would enhance the understanding of how climate-smart agriculture can be tailored to diverse environments and cultural practices, the authors should please revise.

2. Still in the introduction section, which describes the challenges faced by smallholder farmers and the potential benefits of climate-smart agriculture, this section lacks a discussion of the role that governance and policy frameworks play in facilitating or hindering the adoption of such technologies, please the authors must pay attention to this gap.

3. The literature review in section two of this manuscript should be merged with introduction section and critically summarise this section.

4. All the Tables presented in this manuscript must be revised based on the standard journal requirements.

5. The authors did not add discussion on 4.1, 4.2, 4.3, 4.4 and 4.5

6. No seen interpretation on 4.6 and bad notations.

7. The grammatical structure and tenses of the manuscript should be improved.

8. Please, ensure that the line spacing in the text is consistent and in line with journal requirement. Check grammatical errors and ensure proper formatting for all text.

7. PLOS authors have the option to publish the peer review history of their article (what does this mean?). If published, this will include your full peer review and any attached files.

Reviewer #2: No

Reviewer #3: No

---

## [Author Response · Author response to Decision Letter 1]

19 Aug 2024

Author Response to Reviewer 

Dear reviewer, we are delighted to express appreciation for your valuable comments and the time you dedicate to provide such a very good feedbacks. We sincerely believe that the comments are constructive, scientific, and more valuable for this work. As a result, we have attempted to address the comments, and the revised sections are highlighted in red. First and foremost, we apologise for any responses that did not satisfy or address your comments.

Comment≠1: The introduction discusses climate change impacts and the need for climate-smart agriculture in Sub-Saharan Africa (SSA) but fails to address the diverse local contexts that influence the adoption and effectiveness of these practices. Different countries and regions face unique climatic challenges, agricultural methods, and socio-economic conditions. Consequently, more nuanced research findings and examples from specific countries or communities would enhance the understanding of how climate-smart agriculture can be tailored to diverse environments and cultural practices, the authors should please revise. 

Our reply: You have excellent insight; it is true that the introduction part only discussed the consequences of climate change and the necessity for climate-smart agriculture adoption. We endeavour to respond to your comment. As a result, we have considered the effectiveness of CSA, as well as how different countries' socioeconomic and distinctive tactics influenced adoption outcomes. As a result, we have collected contemporary literatures that support the effectiveness, and different elements influence CSA like “In Africa, CSA was embraced as a strategy to address concerns about agricultural productivity, improve adaptation measures, and increase climate resilience. However, in most countries, it has been challenged by a lack of national Climate-Smart Agriculture Investment Plans (CSAIPs) [42], a lack of clear conceptual understanding, inadequate policies, and insufficient capital [43], [44]. It has also been affected by high initial investment costs, labour requirements, and management intensity associated with conservation agriculture and rainwater harvesting [45], gender gaps, ecological and environmental factors [46], [47], shortage of cropland, land tenure issues, lack of adequate knowledge about CSA, slow return on investment ,and insufficient policy and implementation schemes [48], [49]. Climate-smart agriculture has been mainstreamed into agricultural development plans through the construction of regional, sub-regional, and national climate change policies and strategies aimed at mitigating climate change and strengthening African people's adaptive capacity [50]. Smallholders face challenges in implementing climate-smart agriculture (CSA) due to unclear duties, insufficient links between administration levels, limited resources, and political intervention [51]”.

Comment≠2: Still in the introduction section, which describes the challenges faced by smallholder farmers and the potential benefits of climate-smart agriculture, this section lacks a discussion of the role that governance and policy frameworks play in facilitating or hindering the adoption of such technologies, please the authors must pay attention to this gap.

Our reply: Of course, the introduction section does not go into detail regarding the challenges and benefits of CSA, as well as how governance and policy frameworks respond to the adoption of climate smart agriculture. By accepting the comments and providing literature that can answer your queries “African governments face political, economic, and governance problems while implementing climate-smart agriculture (CSA) to achieve the United Nations Sustainable Development Goals [52]. In most African countries, there is a lack of consideration for the replacement, complementing, or conditional effects of policy initiatives on the adoption of smart climate practices [53].The government and policy alternatives for climate smart adoption have inadequate and ineffective interventions in decision making, accessing inputs, and formulating and implementing policies in Africa [54].”

Comment≠3: The literature review in section two of this manuscript should be merged with introduction section and critically summarise this section.

Our reply: Dear Reviewer, We investigated the published articles and found that they had a literature review parts. We have accepted your comment. Because we recognise that your idea of constructing an introduction by incorporating literature into the introductory part will undoubtedly boost the manuscript's quality. Of course, creating a sensible opening that clearly demonstrates the study's limitations would boost its quality. As a result, we rewrote and thoroughly summarised the introduction and literature parts together. Track modification highlights the introduction's summarising and combined sections.

Comment≠4: All the Tables presented in this manuscript must be revised based on the standard journal requirements.

Our reply: We attempted to rewrite the tables, figures, and equations based on the manuscript standard or guideline. We have also referred the published articles in the PLOS one.

Comment≠5: The authors did not add discussion on 4.1, 4.2, 4.3, 4.4 and 4.5

Our reply: We have added the discussion part for 4.1 like “Results show that, with homogeneous technology, the estimated mean technical efficiency of 38.2% implies that, in Africa, almost 62% of the potential agricultural output is untapped [78]. Cereal production in Sub-Saharan Africa has experienced low and unpredictable growth rates due to its rain- fed nature and subsistence orientation over the last four decades [96]. In recent years, crops were planted on 98.6 million hectares, yielding 162 million tons [97].”

For 4.2 “ In LDCs, the impacts of agricultural inputs are stumpy in influencing the growth of agriculture, but producers use cropping intensity to compensate for the dropping average yield per harvest and to assure sustainable land productivity development[59]. Organic fertilizer treatments resulted in an 11% -13% yield increase and a 4% -5% higher net economic gain [67], [68]. After the 2000s, the average annual growth rate of agricultural productivity is expected to be 3.13% for the sample counties [4].”

For 4.3 “With the expansion of technology, CO2 increases world yields by roughly 1.8% every decade while decreasing them by 1.5% every decade [11]. Carbon dioxide (CO2), nitrogen oxide (N2O), and methane (CH4) are the most prevalent greenhouse gases (GHGS), and they are creating a variety of serious consequences in the agriculture sector [12], [13].In this study, climate change and cereal yields show a consistent and similar pattern in the majority of the region from 2005 to 2020. However, numerous Sub-Saharan African countries, including South Africa and Angola, have high carbon emissions and low cereal crop yields; in particular, South Africa's carbon emissions are disproportionately high in relation to its cereal crop yields.”

For 4.4 and 4.5 With a technological efficiency of 38.2%, Africa is predicted to miss out on around 62% of its agricultural potential [78], particularly in Ethiopia, where the estimated TE of 66% implies possibilities for greater crop production efficiency [79]. A firm is considered technically efficient if it is able to manufacture more goods without expanding the number of production inputs. We have investigated and estimated the technical efficiency score of cereal crop production in the region. The results in Fig. 4 demonstrate that the region has an average minimum technical efficiency score of 0.683 and a maximum technical efficiency score of 0.994. This signifies that the region is highly technologically efficient.” “From 1960 to 1991, crop output in SSA increased by 2.7% each year, while food production increased by 2.4% per year. But, worker productivity declined by an average of one percent per year in SSA agriculture [98]. Crop productivity dropped between 2008 and 2019, with no evidence of improvement [99]. “Table 2” shows the average cereal crop yield per unit of input. The results confirmed that the average cereal crop yield per hectare from 2005 to 2020 was 1497.332kg. During these years, 99% of the countries in the region have produced a maximum of 7541kg per hectare. However, the productivity of inputs such as labour, land, and fertilizer use has had different effects on cereal crop yields across countries in the region.”

Comment≠6: No seen interpretation on 4.6 and bad notations.

Our reply: This section is only required for to identify does the variable is stationary or non-stationary. The results are confirmed the stationary of the data. To solve its bad notation we have stated this table in the appendix section.

Comment≠7: The grammatical structure and tenses of the manuscript should be improved.

Our reply: Of course, the manuscript includes grammatical structure issues, and we attempted to resolve these issues by reviewing and rewriting the entire section. Furthermore, our senior staffs, who have experience in publishing manuscripts, have attempted to revise and fix the grammatical errors. 

Comment≠7: Please, ensure that the line spacing in the text is consistent and in line with journal requirement. Check grammatical errors and ensure proper formatting for all text.

Our reply: We rereading and revised very well to ensure the line spacing and text consistency problem. As far as possible, we try to prove the formatting, spacing and other required requirements.

If there is any more please do note hesitate to give

---

## [Decision Letter · Decision Letter 2]

4 Sep 2024

PONE-D-24-08690R2Factor productivity impacts of climate change and estimating the technical efficiency of cereal cropyields: Evidence from sub-Saharan African countriesPLOS ONE

Dear Dr. ALEMU,

Thank you for submitting your manuscript to PLOS ONE. After careful consideration, we feel that it has merit but does not fully meet PLOS ONE’s publication criteria as it currently stands. Therefore, we invite you to submit a revised version of the manuscript that addresses the points raised during the review process.

We look forward to receiving your revised manuscript.

Kind regards,

Fabien MUHIRWA

Academic Editor

PLOS ONE

Journal Requirements:

Reviewers' comments:

Reviewer's Responses to Questions

**Comments to the Author**

1. If the authors have adequately addressed your comments raised in a previous round of review and you feel that this manuscript is now acceptable for publication, you may indicate that here to bypass the “Comments to the Author” section, enter your conflict of interest statement in the “Confidential to Editor” section, and submit your "Accept" recommendation.

Reviewer #3: All comments have been addressed

2. Is the manuscript technically sound, and do the data support the conclusions?

Reviewer #3: Yes

3. Has the statistical analysis been performed appropriately and rigorously? 

Reviewer #3: Yes

4. Have the authors made all data underlying the findings in their manuscript fully available?

Reviewer #3: Yes

5. Is the manuscript presented in an intelligible fashion and written in standard English?

Reviewer #3: Yes

6. Review Comments to the Author

Reviewer #3: This manuscript titled “Factor productivity impacts of climate change and estimating the technical efficiency of cereal crop yields: Evidence from sub-Saharan African countries” estimates the technical efficiency of cereal crop yields and examines the effects of climate change on agricultural productivity in 35 sub-Saharan African countries from 2005 to 2020. Using pooled OLS and stochastic frontier models, the average efficiency score for cereal production was found to be 83%. Labour and fertilizer significantly enhanced efficiency, contributing 51.5% and 5.7%, respectively, while arable land per hectare negatively impacted efficiency by 44.7%. Climate change indicators such as CO2 and methane emissions significantly reduced the productivity of land, labour, and fertilizers, whereas GDP per capita positively influenced overall agricultural productivity. The findings highlight high technical efficiency in the region, primarily driven by labour and fertilizer inputs, but these have been adversely affected by climate change. This work is an interesting contribution, and the authors are commended for their efforts.

I wholeheartedly recommend this paper for publication with no revisions.

7. PLOS authors have the option to publish the peer review history of their article (what does this mean?). If published, this will include your full peer review and any attached files.

Reviewer #3: No

---

## [Author Response · Author response to Decision Letter 2]

7 Sep 2024

Author Response letter to Editor

Dear editor of the PLOS ONE, we would like to thank you for your unreserved contribution. Your comments were constructive this work from the first round to until it reaches at this stages. We understand your aid is how much valuable to improve the scientific value of this manuscript. The referencing is cited using mendeley software, but it is not free from some referencing problem. As you request us to edit and make a correction on references, we try to reread, edited and replaced some old references by recent references. While we are editing the reference lists, there is problem of incomplete referencing that raised by software problem as well as use of old references. But there is no referenced source that is not listed in bibliography because all cited references are listed in the reference list section. Some retracted references replaced by recent references like “ 1) F. Muhirwa et al., “Ecological balance emerges in implementing the water-energy-food security nexus in well-developed countries in Africa,” Sci. Total Environ., vol. 833, 2022, doi: 10.1016/j.scitotenv.2022.154999.2) Y. Karavias and E. Tzavalis, “Generalized fixed-T panel unit root tests,” Scand. J. Stat., vol. 46, no. 4, pp. 1227–1251, 2019, doi: 10.1111/sjos.12392.

---

## [Editor Report · Decision Letter 3]

11 Sep 2024

Factor productivity impacts of climate change and estimating the technical efficiency of cereal cropyields: Evidence from sub-Saharan African countries

PONE-D-24-08690R3

Dear Dr. ALEMU,

We’re pleased to inform you that your manuscript has been judged scientifically suitable for publication and will be formally accepted for publication once it meets all outstanding technical requirements.

Kind regards,

Fabien MUHIRWA

Academic Editor

PLOS ONE
---

## [Editor Report · Acceptance letter]

18 Sep 2024

PONE-D-24-08690R3 

PLOS ONE

Dear Dr. ALEMU, 

I'm pleased to inform you that your manuscript has been deemed suitable for publication in PLOS ONE. Congratulations! Your manuscript is now being handed over to our production team.

Kind regards, 

on behalf of

Dr. Fabien MUHIRWA 

Academic Editor

PLOS ONE